# Security Enhancement for Deep Reinforcement Learning-Based Strategy in Energy-Efficient Wireless Sensor Networks

**DOI:** 10.3390/s24061993

**Published:** 2024-03-21

**Authors:** Liyazhou Hu, Chao Han, Xiaojun Wang, Han Zhu, Jian Ouyang

**Affiliations:** 1School of Computer Science and Engineering, Macau University of Science and Technology, Macau 999078, China; liyazhouhu@ieee.org; 2Industrial Training Center, Guangdong Polytechnic Normal University, Guangzhou 510665, China; wxj65881010@163.com; 3China Mobile Jianshe Co., Ltd. Zhejiang Branch, Hangzhou 310013, China; hanchao1@cmtt.chinamobile.com; 4Faculty of Applied Sciences, Macao Polytechnic University, Macau 999078, China

**Keywords:** deep neural network (DNN), deep reinforcement learning (DRL), energy efficiency, security, wireless sensor networks (WSNs)

## Abstract

Energy efficiency and security issues are the main concerns in wireless sensor networks (WSNs) because of limited energy resources and the broadcast nature of wireless communication. Therefore, how to improve the energy efficiency of WSNs while enhancing security performance has attracted widespread attention. In order to solve this problem, this paper proposes a new deep reinforcement learning (DRL)-based strategy, i.e., DeepNR strategy, to enhance the energy efficiency and security performance of WSN. Specifically, the proposed DeepNR strategy approximates the Q-value by designing a deep neural network (DNN) to adaptively learn the state information. It also designs DRL-based multi-level decision-making to learn and optimize the data transmission paths in real time, which eventually achieves accurate prediction and decision-making of the network. To further enhance security performance, the DeepNR strategy includes a defense mechanism that responds to detected attacks in real time to ensure the normal operation of the network. In addition, DeepNR adaptively adjusts its strategy to cope with changing network environments and attack patterns through deep learning models. Experimental results show that the proposed DeepNR outperforms the conventional methods, demonstrating a remarkable 30% improvement in network lifespan, a 25% increase in network data throughput, and a 20% enhancement in security measures.

## 1. Introduction

With the development of wireless communication technology, digital signal processing, and microelectronics, wireless sensor networks (WSNs) are gradually transforming from theoretical research to practical applications in the Internet of Things (IoT). For example, lighting control, security monitoring, and air-conditioning control in the smart home are using WSNs to achieve home automation control [1]. Moreover, WSNs are also widely used in various fields, such as medical IoT [2], environmental monitoring, agriculture, military reconnaissance [3], and traffic [4].

WSN is a distributed network consisting of a large number of sensor nodes freely combined to monitor physical or environmental conditions, such as temperature, sound, pressure, etc., and to cooperatively pass their data through the network to a main location [5,6]. Each node can sense, process, and transmit information. WSNs are characterized by their use of a wireless medium for communication and their reliance on sensor nodes, which are typically battery-powered and limited in computation and storage capabilities. These networks utilize various communication protocols designed to conserve energy, a critical resource in WSNs. Techniques such as data aggregation [7], where data from multiple sensors are combined to reduce the number of transmissions, and duty cycling [8], where nodes toggle between active and sleep states to save energy, are commonly employed to extend the network’s lifetime. Additionally, WSNs employ routing techniques, like low energy adaptive clustering hierarchy (LEACH) [9] and power-efficient gathering in sensor information systems (PEGASIS) [10], to enhance energy efficiency. These protocols aim to minimize energy consumption during data transmission, a significant challenge due to the limited battery life of sensor nodes. Despite the advancements in WSN technologies, several challenges remain [11,12]. Firstly, energy efficiency continues to be a paramount concern, as the finite energy resources of sensor nodes limit the network’s operational lifetime. Secondly, security in WSNs is a critical issue due to the sensitive nature of the data collected and the potential for malicious attacks on these networks. Challenges such as data interception, unauthorized access, and node compromise need robust security mechanisms to ensure data integrity and network resilience. Furthermore, scalability and adaptability pose significant challenges as WSNs are deployed in increasingly complex and dynamic environments [13]. Ensuring that WSNs can scale to accommodate large numbers of sensor nodes and adapt to changing environmental conditions without compromising performance or energy efficiency requires sophisticated network design and management strategies. Therefore, ensuring the efficient transmission of wireless communication and achieving good security performance are critical issues in WSN research.

In recent years, the rapid development of deep learning and reinforcement learning techniques has significantly impacted various industries, including healthcare, autonomous vehicles, and smart grid management [14,15]. These advancements have led to the emergence of deep reinforcement learning (DRL), a powerful tool that combines the representational ability of deep learning with the decision-making prowess of reinforcement learning [16]. DRL excels in environments where the acquisition of an optimal policy requires the understanding of complex inputs and the making of sequential decisions, making it particularly suited for applications in WSNs. To address the above challenges, this paper proposes a novel DeepNR strategy, leveraging DRL to enhance both the energy efficiency and security of WSNs. By integrating advanced DRL algorithms with traditional WSN operations, the proposed DeepNR strategy not only addresses the aforementioned challenges but also opens new avenues for the development of intelligent and sustainable WSNs. Compared with traditional decision-making, the proposed DeepNR can more accurately predict network status and external threats and, thus, make more reasonable decisions. The main contributions of this paper can be summarized as follows:We propose a novel DeppNR strategy that intelligently optimizes energy consumption in WSNs, significantly extending network lifecycles beyond current benchmarks.The proposed DeepNR strategy introduces cutting-edge security enhancements directly embedded within the DRL model, offering robust protection against a wide array of cyber threats, thereby elevating the network’s resilience.The experiments verify the effectiveness and superiority of the proposed DeepNR. Compared with other conventional methods, DeepNR can significantly improve the energy efficiency of WSN while ensuring security.

The remainder of the paper is organized as follows: Section 2 presents the current research related to the energy efficiency and security of WSN. The proposed methods are described in Section 3. The experimental results are shown in Section 4. Finally, Section 5 concludes this paper.

## 2. Related Work

### 2.1. Traditional Solutions

Traditional methods for improving energy efficiency in WSNs often involve optimizing communication protocols and employing low-power operational modes. For instance, techniques like data aggregation, dynamic power adjustment, and duty cycling have been widely explored. Perrig [17] proposed data aggregation, dynamic adjustment of transmission power, and low-power monitoring schemes to improve energy efficiency. Seo [18] addressed the problem of minimizing the energy consumption of duty-cycled sensors, using the optimal sleep interval as a closed solution to minimize energy consumption. On the security front, encryption, authentication, and intrusion detection mechanisms form the core of most strategies aimed at safeguarding network communications. Akoglu [19] proposed encrypted communication, authentication, and intrusion detection methods in network communication security policies. Ioannou [20] used a binary logistic regression statistical tool to categorize local sensor activity as benign or malicious, effectively detecting malicious activity in the 88–100% range. However, these approaches only consider a one-sided increment in performance benefits, often at the cost of other performances. When low-power communication and computation strategies are used to improve energy efficiency, network security is reduced, making the network more vulnerable to external attacks. Therefore, it is crucial to achieve compatibility between energy efficiency and security in WSNs.

### 2.2. Machine Learning-Based Solutions

A neural network is an algorithm that performs calculations by imitating the connections and communication between neurons in the human brain. It consists of multiple layers of neurons, each of which processes input data by adjusting weights and activation functions [21]. The performance of deep neural networks (DNNs) in areas such as image recognition, speech recognition, and natural language processing is superior to that of traditional methods, which provides an opportunity for the development of WSNs. In [22], the authors proposed an energy-aware routing scheme based on a multi-objective cluster head for secure data aggregation in WSN. The proposed scheme outperformed the existing models in terms of network lifespan, packet transmission rate, and delay. The authors of [23] developed a deep learning-based graph neural network and proposed a hybrid fixed-variant search method to conserve the energy of the sensors to extend the network lifespan of the WSN. In [24], the authors used DNN techniques in a WSN system for IoT-based applications, where customized deep learning techniques could detect intrusions in real time accurately. The authors of [25] introduced DNN algorithms into WSNs to promote link reliability in WSNs, where DNN algorithms were used to evaluate input parameters such as received bandwidth and delay. Meanwhile, reinforcement learning has emerged as a learning method that improves the performance of an agent through its interaction with the environment. The agent optimizes its behavior strategy by observing the state of the environment, performing corresponding actions, and receiving reward signals [26]. In other words, reinforcement learning [27] is how an agent learns the best strategy to maximize the expected cumulative reward by interacting with the environment. The authors of [28] proposed a hierarchical edge caching architecture for vehicular networking using distributed multi-agent reinforcement learning, which greatly improved the edge hit rate. Common reinforcement learning methods include value-based approaches, policy-based approaches, and actor–critic approaches. Policy-based approaches are complementary to value-based approaches, while actor–critic approaches are a combination of the first two. Reinforcement learning follows the Markov Decision Process (MDP) [29]. Only if the training process conforms to the MDP can the next state be inferred based on the current state and the action taken.

DRL combines DNN and reinforcement learning by using a DNN to represent a policy or value function for more accurate prediction and decision-making and then optimizes the model parameters using the reinforcement learning policy for better performance. The DRL method has already played an indispensable role in the WSN field. In WSN routing algorithms [30], the authors used deep learning models to learn the network state in real time and dynamically select the best routing path based on the network state. In WSN energy management [31], a WSN consisting of multiple local subnets was proposed, where DRL constructs a MDP model for the optimization problem in the subnetwork to find the optimal resource allocation strategy. In the field of WSN security [32], reconfigurable intelligent surfaces (RISs) are used to assist secure WSNs by maximizing the secrecy rate through the joint optimization of RIS phase shifts, sensor scheduling policies, and interference power. However, achieving the optimal trade-off between energy efficiency and security for WSNs is still a concern.

### 2.3. Existing Challenges and Gaps

The deployment and operation of WSNs are fraught with challenges, notably in balancing energy efficiency with robust security measures. Traditional approaches often tackle these issues in isolation, leading to sub-optimal network performance. Key challenges include the following:Energy Efficiency. Given the limited power resources of sensor nodes, extending network lifespan while maintaining operational effectiveness remains a primary concern.Security. The open nature of wireless communication exposes WSNs to a range of security threats, from data breaches to node tampering, necessitating comprehensive security solutions.Adaptability. Dynamic network conditions and evolving threat landscapes require adaptable strategies that traditional static methods cannot provide.

Our proposed DeepNR strategy leverages DRL to address these challenges in an integrated manner. Unlike conventional methods, DeepNR dynamically optimizes both energy efficiency and security, adapting to changing network conditions and security threats in real time. This approach not only mitigates the limitations of existing solutions but also introduces a scalable, flexible framework for the comprehensive management of WSNs.

## 3. Proposed Method

In this section, we describe the proposed DeepNR strategy that uses DNN to represent the Q-value function and DRL to optimize network strategy.

### 3.1. Network Design

To achieve our proposed network strategy, we have implemented a deep Q-network (DQN) model, a subset of DRL known for its robustness in handling complex, high-dimensional environments. The DQN model integrates the advanced capability of deep neural networks with the strategic decision-making framework of Q-learning, making it particularly suited for the dynamic and challenging conditions of WSNs. Our DQN model is designed with a custom neural network architecture, comprising an input layer that captures the state of the WSN, several hidden layers that process this information, and an output layer that predicts the best possible actions to optimize both energy efficiency and security. This model allows for a nuanced understanding and interaction with the network environment, facilitating real-time adaptations to changing conditions and threats.

Figure 1 represents the neural network architecture of the proposed DeepNR strategy. The neural network is designed to process the state of the network and output a probability distribution of potential actions to be taken. Specifically, the state *s* is an input to the neural network representing the current condition of the WSN. It encapsulates various parameters such as sensor node energy levels, data transmission rates, and the presence of potential security threats. In the context of our DeepNR strategy, the state can include a wide array of sensor data, network metrics, and environmental factors that influence the decision-making process. The output of the neural network is a set of actions represented by a probability distribution. Each action corresponds to a possible decision or operation that can be performed by the WSN, such as routing adjustments, power management decisions, or initiating security protocols. The probability distribution indicates the likelihood of each action being the optimal choice given the current state of the network. The input layer contains 3*N* + *T* × (3*N* + 2*N*) neurons. Hidden layers 1 and 2 contain 256 neurons and 128 neurons, respectively, with the ReLU activation function. The output layer has 2*N* neurons representing the Q-value of the action vector. Moreover, we represent the policy π using a DNN.

The input of the network is *x* = [*s*, $h_{1}$, $h_{2}$, …, $h_{T}$]. *s* is the current state, which contains the energy information, communication quality, and recent security events of each node. *h* is the historical status of the node, including the actions and status of each node in the previous time step. In this paper, we use experience replay and target network techniques to train our DNN. Experience replay allows the network to learn from past experiences, while the target network helps stabilize the learning process. The DNN is used to approximate the Q-value function, replacing the traditional Q-learning algorithm [33].

To sum up, a neural network acts as an approximator of the Q-value function in our proposed DeepNR strategy, while Q-learning is a framework for finding the optimal strategy by iteratively updating the Q-value. By combining neural networks and Q-learning, the DeepNR strategy is able to optimize the network performance in a complex WSN environment. Compared with DNN, the innovative aspects of our approach are highlighted in several key areas:Customization for WSNs. We have designed the DNN architecture specifically to handle the high-dimensional state space characteristic of WSNs, which is not addressed by standard DNN applications.Real-time adaptation. The combination of DNNs with DRL enables our strategy to learn and adapt in real time to changing network conditions and attack patterns, a critical requirement for WSNs that is not commonly met by traditional DNN applications.Multi-level decision-making. Our approach uses a DNN-based policy to perform multi-level decision-making, dynamically adjusting network strategies to balance energy efficiency with robust security measures.Composite reward function. We introduce a new composite reward function within the DRL framework to guide the training process of the DNN, which is specifically designed to address the dual objectives of energy efficiency and security in WSNs.

### 3.2. Strategy Optimization

WSNs play a pivotal role in numerous applications, ranging from environmental monitoring to enabling smart cities, and face the dual challenge of limited energy resources and vulnerability to security threats. Traditional methods often address these challenges in isolation, leading to compromises that can weaken network performance. The proposed DeepNR strategy is motivated by the vision of a unified solution that leverages DRL to dynamically optimize network operations, thereby achieving a balanced improvement in energy efficiency and security without compromising on either front.

In our proposed strategy, it is crucial to model the communication quality accurately between nodes. This model provides the basis for not only our energy efficiency and security strategies but also the overall performance optimization of the network.

Given the wireless communication characteristics of WSNs, the communication quality between node *i* and node *j* can be expressed as
(1)$$Q_{ij}=\frac{P_{tx}\times G_{tx}\times G_{rx}}{L\left(d_{ij}\right)\times V}$$
where $P_{tx}$ is the transmission power and $G_{tx}$ and $G_{rx}$ are the transmitting and receiving antenna gains, respectively. $L\left(d_{ij}\right)$ represents the path loss function, and *V* is the noise power.

Considering the energy consumption of each node and the security threats of WSN, we define an energy efficiency index and a security index:(2)$$E_{eff}\left(i\right)=\frac{D_{i}}{E_{i}}$$
(3)$$S_{risk}\left(i\right)=\frac{A_{i}}{T}$$
where $D_{i}$ is the data throughput of node *i*. $E_{i}$ is the energy consumption of node *i* and can be calculated by (4). $A_{i}$ is the number of attacks that node *i* received within the time window *T*.
(4)$$E_{i}=P_{tx}\times t_{tx}+P_{rx}\times t_{rx}+P_{idle}\times t_{idle}$$
where $P_{tx},P_{rx}$, and $P_{idle}$ represent the power of the node in transmit, receive, and idle states, respectively. Meanwhile, $t_{tx},t_{rx}$, and $t_{idle}$ are the durations spent by the node in these states, respectively.

In order to maximize the energy efficiency and minimize the security risk of the entire network, we define the optimization problem as follows:(5)$$\begin{array}{c} \max_{p}\sum_{i=1}^{N}E_{eff}\left(i\right)=\max_{p}\frac{D_{i}}{E_{i}} \end{array}$$
(6)$$\begin{array}{c} \min_{p}\sum_{i=1}^{N}S_{risk}\left(i\right)=\max_{p}\frac{A_{i}}{T} \end{array}$$
where *p* is the policy of the network, including the transmission power and encryption level of each node.

In order to use DRL to optimize the strategy of WSN, we first define a MDP, which can be represented as a quintuple array (S,A,P,R,γ). *S* is the state space. *A* is the action space. *P* is the state transfer probability, defined as $P\left(s^{\prime }|s,a\right)$. *R* is the reward function, defined as $R\left(s,a,s^{\prime }\right)$. γ is the discount factor. To consider both energy efficiency and security performance, we employ DRL to learn a strategy that optimizes the performance of the entire network. We define a new composite reward function as
(7)$$R\left(s,a\right)=\alpha \times \sum_{i=1}^{N}E_{eff}\left(i\right)-\beta \times \sum_{i=1}^{N}S_{risk}\left(i\right)$$
where α and β are the weighting parameters. The goal of reinforcement learning is to maximize the expected reward of the network. It can be expressed as
(8)$$J\left(\pi \right)=E_{\pi }\left(R_{1}+\gamma R_{2}+\gamma^{2}R_{3}+\ldots \right]$$
where $R_{t}$ is the reward at time *t*. In order to prove that our proposed DRL-based neural network structure rather than Q-value is the optimal strategy, we utilize Bellman’s optimality principle for mathematical verification [34]. The Q-value function generated by the neural network satisfies the following Bellman equation:(9)$$Q\left(s,a\right)=R\left(s,a\right)+\gamma \times E\left[\max_{a^{\prime }}Q\left(s^{\prime },a^{\prime }\right)\right]$$
where γ is the discount factor, $s^{\prime }$ denotes the subsequent state, and $a^{\prime }$ represents the subsequent action.

The goal of strategy optimization is to find the policy π that maximizes the Q-value function. This can be expressed in the following optimization problem:(10)$$\max_{\pi }E\left[Q\left(s,a\right)\right]$$

Utilizing the policy gradient method, we can determine the gradient of the policy as
(11)∇J(π)=E[∇logπ(a∣s)(Q(s,a)−V(s))]
where V(s) is the value function of state *s*. By iteratively updating the policy parameters, we can derive the optimal policy.

Note that, unlike deep Q-learning, our proposed approach aims to create a more robust and effective strategy optimization process for WSNs, ensuring that DeepNR can handle the intricacies and challenges specific to these networks. This includes but is not limited to the diverse and dynamic nature of network topologies, the variable energy profiles of sensor nodes, and the evolving threat landscape in terms of network security. Specifically, our optimization process has the following characteristics:Enhanced state space representation. The state space in DeepNR has been designed to encapsulate a more comprehensive set of network parameters, which are specifically chosen to represent the complex dynamics of WSNs.Customized reward function. We employ a customized reward function that is particularly formulated to address the dual objectives of maximizing energy efficiency and ensuring network security, which is not traditionally the focus of standard deep Q-learning applications.Policy optimization for WSNs. Our approach adapts the policy optimization process to suit the specific operational constraints and performance goals of WSNs, such as node energy limitations and the need for rapid response to security threats.Advanced experience replay mechanism. We have implemented an advanced experience replay mechanism that better suits the temporal and spatial variability in WSNs, enhancing the learning process beyond the typical deep Q-learning approach.Integration with WSN-specific protocols. DeepNR is integrated with WSN-specific protocols, which enable a seamless transition from learned strategies to actionable policies in a real-world network environment.

## 4. Experiment and Evaluation

To verify the actual performance of the proposed DeepNR strategy, our test environment was designed to replicate a typical WSN deployment, utilizing 500 MICAz sensor nodes equipped with CC2420 communication modules, which were randomly deployed in a 500 m × 500 m area. Each node had a communication range of 50 m and an initial energy of 100 J. All nodes ran on the TinyOS operating system. We configured the nodes to operate within an IEEE 802.15.4 standard network.

The network comprised multiple PANs, each with a designated full-function device (FFD) serving as the PAN coordinator. The remaining nodes were designated as reduced-function devices (RFDs), acting as end nodes with specific sensing and communication duties. The nodes were organized into hierarchical levels to facilitate efficient data routing and aggregation. The number of hierarchy levels was determined based on the network size and density, with specific attention to optimizing the trade-off between energy consumption and latency. To ensure a comprehensive evaluation of the proposed DeepNR strategy, we executed a total of 100 tests for each scenario to ensure statistical significance in our results. The duration of each test was set to simulate real-world operation over a period of 24 h, which allowed us to observe the performance and resilience of the proposed strategy under sustained operation. Note that the test duration should be long enough to observe the behavior of the network under stress but not excessively long to deplete energy resources unrealistically. Moreover, our experimental setup was carefully calibrated to ensure a realistic simulation of WSN operations, taking into account the typical energy constraints and security requirements of such networks.

In order to fairly evaluate the performance of the DeepNR strategy, we selected three WSN strategies, LEACH (a classic clustering routing algorithm) [35], PEGASIS (a link-based data fusion strategy) [36], and DEEC (an energy-based clustering algorithm) [37], as baseline algorithms.

### 4.1. Network Life Cycle

In WSNs with various node number sizes, we could compare their life cycles based on the total energy consumption of the network. The number of nodes was set as 100, 200, 300, 400, and 500, respectively.

As shown in Figure 2, the horizontal axis represents the network size (the number of nodes), and the vertical axis represents the total network energy consumption (J). The total energy consumption is the sum of all energy expenditures by the nodes within the wireless sensor network over the entire duration of the simulation. This metric encompasses the energy utilized for various network activities, including transmission energy (the energy consumed by a node to transmit data packets to other nodes or the base station), reception energy (the energy used by a node to receive data packets from other nodes), processing energy (the energy required for computational tasks by the nodes, such as data aggregation and protocol computations), and idle energy (the baseline energy consumption when nodes are not actively transmitting or receiving but are maintaining an operational state ready to engage in network activities). Obviously, as the network size increases, the network energy consumption also increases. Moreover, our proposed DeepNR strategy effectively reduces the energy consumption in wireless sensing and extends the network life cycle compared to other strategies, which can be attributed to it optimizing node communication and data processing, thereby reducing the energy expended per node.

The DeepNR strategy optimizes node communication by selecting the most energy-efficient data transmission paths based on the current network state, which is dynamically learned by the DRL model. This optimization reduces the overall energy consumption, allowing nodes to conserve energy and extend the operational period of the network. Furthermore, the DeepNR strategy enhances data processing efficiency by employing intelligent data aggregation techniques that minimize processing overhead, thereby preserving the nodes’ battery lives. The adaptive learning aspect of the DRL model continuously refines these processes in real time, ensuring that the network remains energy-efficient even as conditions change. The DeepNR strategy employs a multi-level decision-making process that intelligently allocates network resources, such as bandwidth and power, to where they are most needed. This not only enhances the network’s energy efficiency but also ensures that critical areas of the network maintain functionality for longer periods, thereby extending the network’s useful life.

### 4.2. Data Throughput

With the continuous development and innovation of the communication environment, the delay standards for wireless sensing communications are also improving. With the increasing demand for real-time data and instant feedback, wireless sensing communication systems need to be able to transmit and process data at a faster speed. In order to accommodate the needs of various communication environments, we simulated the average communication delays for four strategies under different network loads (packets per second).

In Figure 3, we examine the average communication delay under various network loads. It is clear that as the network load increases, the communication latency for each strategy increases accordingly. It is noteworthy that DeepNR consistently exhibits low communication latency across all network loads, which can be attributed to its efficient data routing algorithms and advanced packet processing mechanisms. These features enable it to manage high data traffic more efficiently, reducing the time packets spend in the queue and transmission, thereby minimizing overall communication latency.

The proposed DeepNR strategy enhances data throughput by employing efficient data routing algorithms and advanced packet processing mechanisms. These algorithms are designed to reduce communication latency by ensuring that data packets take the most energy-efficient and least congested paths through the network. This not only speeds up the transmission time but also reduces the likelihood of packet loss and the need for retransmissions, which can significantly slow down network throughput. Moreover, our strategy incorporates intelligent packet processing techniques that prioritize data packets based on their importance and time sensitivity. By processing and forwarding packets in a manner that reflects their urgency, DeepNR ensures that critical data are transmitted rapidly, enhancing the overall throughput of the network. These mechanisms are underpinned by the DRL model within DeepNR, which learns from the network environment and adapts the routing and processing strategies in real time. This allows for a responsive and agile network that can maintain high throughput even under varying load conditions.

### 4.3. Security Assessment

To evaluate the security performance of the proposed DeepNR strategy, we simulated three common attack models: single-node attacks, distributed denial-of-service (DDoS) attacks, and deception attacks. We then compared the transmission performance of networks under different strategies and assessed the robustness based on the network lifecycle under various node failure rates.

Our DeepNR strategy incorporates advanced detection mechanisms that utilize the predictive power of DRL to identify potential security threats before they can impact the network. This proactive stance allows for immediate and effective response measures, reducing the likelihood of successful attacks and, consequently, the number of affected nodes. The advanced detection capabilities of DeepNR are designed to identify threats in real time, utilizing a combination of anomaly detection algorithms and pattern recognition trained through DRL. Upon detecting a threat, DeepNR initiates a pre-determined response protocol, which may include re-routing data, isolating affected nodes, or altering transmission patterns to minimize the attack’s impact. DeepNR’s DRL model allows the network to adapt its security measures dynamically. By continuously learning from network interactions and detected threats, it becomes increasingly effective in predicting and preventing attacks. This ongoing learning process means that the network’s security measures evolve in response to the changing tactics of attackers, ensuring that the system remains robust against both known and novel threats.

The number of affected nodes under each attack model is shown in Figure 4. The performances of various strategies against various malicious attacks in decreasing order are as follows: DeepNR, DEEC, PEGASIS, LEACH. This result shows the superiority of our proposed DeepNR strategy in terms of security performance. DeepNR consistently shows fewer affected nodes across a wide range of attack models, demonstrating not only its resilience but also its advanced security protocols. This is due to the fact that DeepNR is equipped with superior mechanisms to detect and mitigate a range of network threats, thus ensuring better network integrity and security compared to other strategies.

Figure 5 shows the network life cycle corresponding to various node failure rates. The ability of a network strategy to maintain its operational lifetime against increasing node failures is critical to ensuring uninterrupted service and reducing maintenance costs. The resilience of the network strategy comes to the forefront when faced with escalating node failures. Under varying node failure rates, the proposed DeepNR continues to show its robustness and outperforms other strategies in terms of the network life cycle. It is closely followed by DEEC and PEGASIS, with LEACH showing the worst performance. However, as the node failure rate rises, the distinction between the four strategies diminishes, which highlights the need for continued research and development to further enhance the ability of networks to cope with various failures.

### 4.4. Energy Distribution and Balance

DeepNR implements an advanced energy management system that monitors the energy levels of each node and redistributes tasks across the network to prevent nodes from depleting their energy reserves prematurely. This system is underpinned by the DRL model, which predicts future energy requirements and adapts the network’s behavior to maintain energy balance. To further demonstrate the performance of the proposed DeepNR strategy, Figure 6 analyses the energy distribution of each node in the network. It is clear that a higher percentage of nodes (90%) consume energy in the range of 40–50 J under the DeepNR strategy compared to the other strategies. In addition, only 2% of the nodes under the DeepNR strategy consume more than 50 J of energy, which is significantly lower than the other strategies. It indicates that the DeepNR strategy is more effective in energy management and ensures that most of the nodes consume energy in a relatively low range. Therefore, concerning the above experimental results, the proposed DeepNR strategy is able to achieve a good balance between security and energy consumption.

The proposed DeepNR strategy includes algorithmic optimizations that dynamically adjust the nodes’ energy consumption based on real-time network demands and the remaining energy levels of the nodes. This approach prevents the over-utilization of nodes that are critical for network connectivity and ensures that energy consumption is more evenly distributed across the network. By maintaining a balanced energy distribution among nodes, the DeepNR strategy not only extends the network’s operational lifetime but also reduces the need for frequent maintenance and node replacements.

## 5. Conclusions

In this paper, a novel DeepNR method is proposed to address the problems of energy efficiency and security in wireless sensor networks. Specifically, we designed a neural network structure to accurately approximate Q-value based on the deep reinforcement learning method, where experience replay and target network techniques are adopted to train our deep neural network. In order to ensure the balance between energy efficiency and security in WSN, we design a multi-level decision-making framework and define a new composite reward function to update the Q-value to obtain the optimal strategy for optimizing network performance. By making decisions based on the information learned from the deep neural network, we can ultimately achieve a balance between the energy efficiency of the nodes and the entire transmission security of networks. The experimental results show that DeepNR can significantly improve the network life cycle, network data throughput, and security compared with other methods.

Considering that modern WSNs may include various sensor types (e.g., temperature, humidity, images, etc.), future work will extend the DeepNR strategy to multimodal networks. Moreover, we intend to explore the application of DeepNR to more sophisticated and AI-driven attack models, including zero-day attacks that present unique challenges to WSN security. With the proliferation of IoT devices, future work will involve extending the DeepNR strategy to encompass a broader range of sensor types and communication protocols inherent to IoT ecosystems. We will delve into cross-layer optimization techniques that account for the interplay between physical, MAC, network, and application layers within WSNs. In addition to DRL mentioned in this paper, many other AI techniques, such as generative adversarial networks and neural symbolic learning, can be integrated with the DeepNR strategy to further enhance its performance.

## Figures and Tables

**Figure 1 sensors-24-01993-f001:**
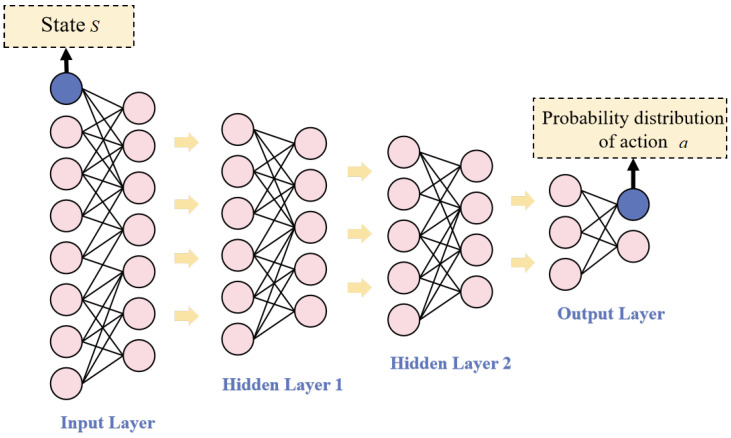
The neural network framework of DeepNR.

**Figure 2 sensors-24-01993-f002:**
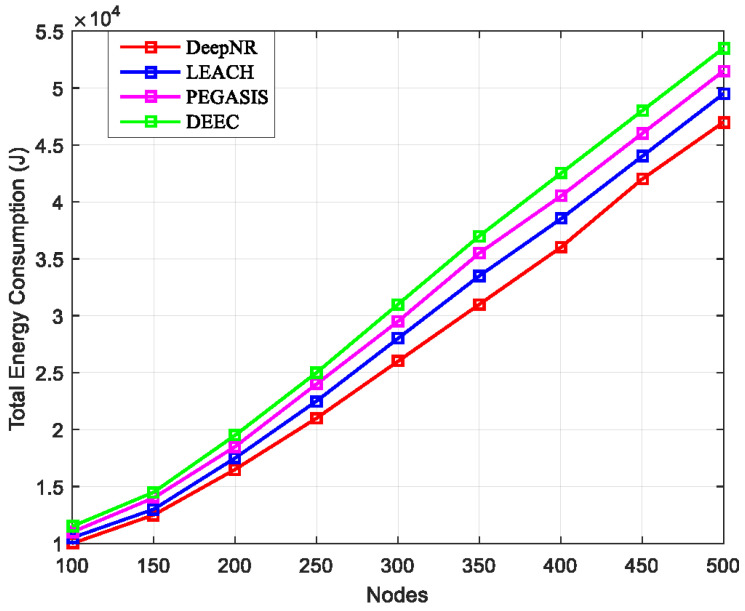
Total energy consumption for four strategies under various numbers of nodes.

**Figure 3 sensors-24-01993-f003:**
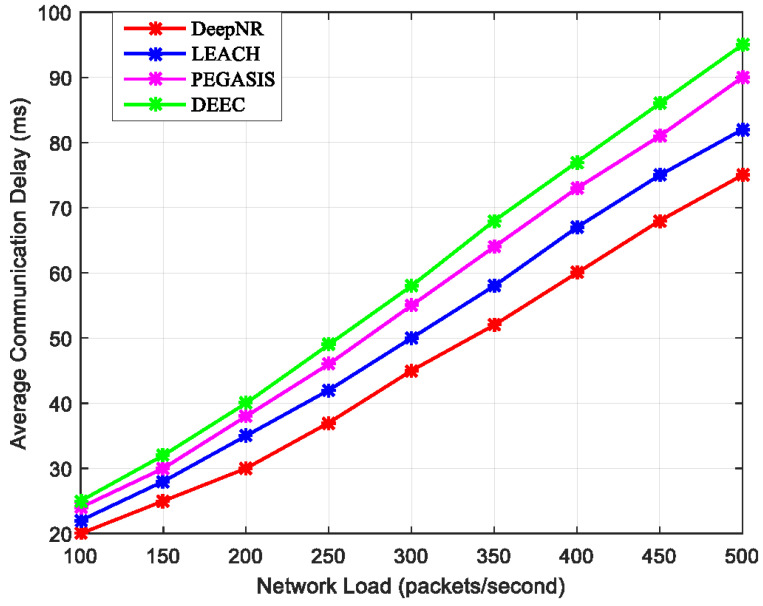
Average communication delays for four strategies under various network loads.

**Figure 4 sensors-24-01993-f004:**
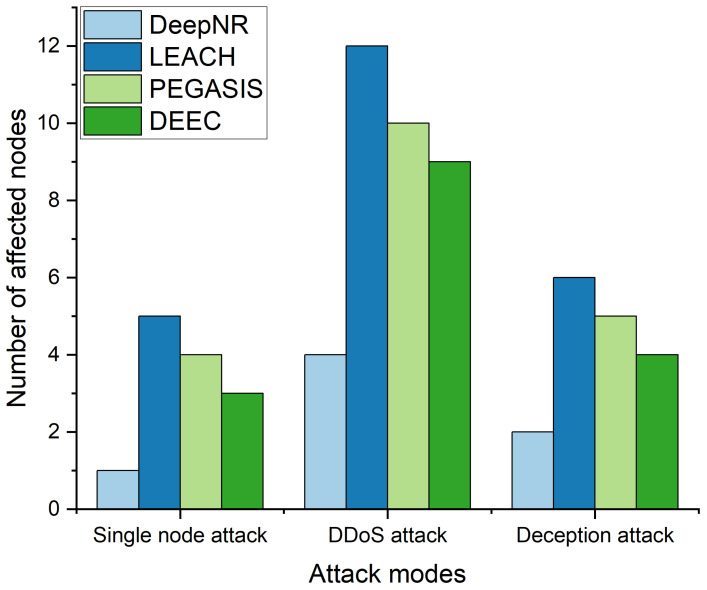
Number of affected nodes for four strategies under various attack models.

**Figure 5 sensors-24-01993-f005:**
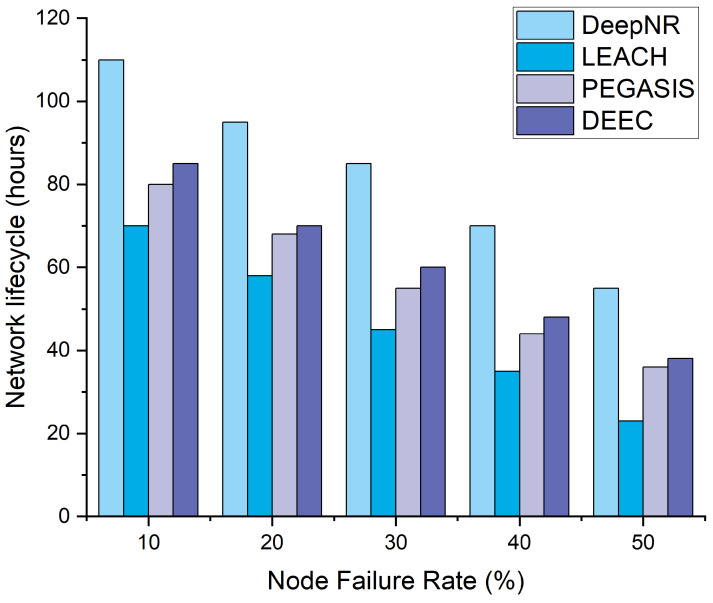
Network lifecycle for four strategies under various node failure rates.

**Figure 6 sensors-24-01993-f006:**
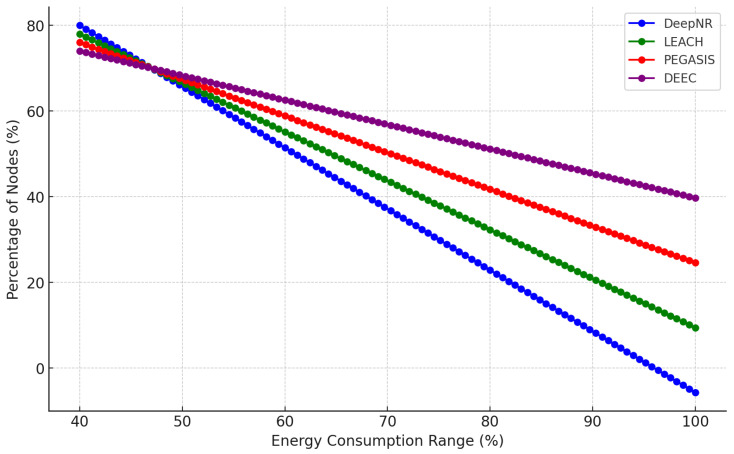
Energy consumption range against node percentage (%) under various strategies.

## Data Availability

The data that support the findings of this study are available from the corresponding author, upon reasonable request. The data are not publicly available due to privacy restrictions.

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
