# Peer review of "Security Enhancement for Deep Reinforcement Learning-Based Strategy in Energy-Efficient Wireless Sensor Networks"

_sensors, 2024, doi:10.3390/s24061993_

Round 1

Reviewer 1 Report

Comments and Suggestions for Authors

It is still a research problem of how to improve the energy efficiency of wireless sensor networks while enhancing security performance. In order to solve this problem, this paper proposes a new deep reinforcement learning (DRL)-based strategy, i.e., DeepNR strategy, to enhance the energy efficiency and security performance of WSN. The proposed DeepNR strategy approximates the Q-value by designing a deep neural network (DNN) to adaptively learn the state information. The proposed solution also designs DRL-based multilevel decision-making to learn and optimize the data transmission paths in real-time. Overall this is a solid study and some comments are as follows.

1. Add more specific numerical results in the abstract to support the effectiveness of the proposed DeepNR strategy.

2. The research background of WSN in the introduction is not enough. The authors should add basic concepts, key techniques and existing challenges in WSN in the introduction section.

3. "In recent years, with the rapid development of deep learning and reinforcement learning techniques in various industries..." This paragraph is not enough for research motivations. The authors should clearly state the research questions and justify why DRL is necessary, instead of existing solutions.

4. The first two points in the summarized contributions are basically the same meaning in the introduction section.

5. The related work section is not well-organized. Different subsections are suggested to use to distinguish between traditional and DRL solutions.

6. The authors should summarize existing research challenges and gaps in the literature and state how they plan to overcome these challenges in the related work section.

7. The latest research progresses should be added, for example, the graph-based deep learning methods for relevant problems with the following references: Graph-based Deep Learning for Communication Networks: A Survey, Multi‐objective cluster head‐based energy aware routing using optimized auto‐metric graph neural network for secured data aggregation in Wireless Sensor Network, Enhancing network lifespan in wireless sensor networks using deep learning based Graph Neural Network and Optimal deployment of large-scale wireless sensor networks based on graph clustering and matrix factorization.

8. The authors should explain the state and action first for Figure 1. The neural network framework of DeepNR.

9. Based on 3.1. Network Design, the authors are basically using a DNN structure, which is very mature. The authors should justify their novelty if a mature technique is used.

10. Based on 3.2. Strategy Optimization, it is difficult to tell the difference between the proposed method and the deep Q-Learning method, which is well-known in the literature.

11. The evaluation metrics in the experiments are not fully defined and explained, for example, how to define total energy consumption in Figure 2.

12. More specific future research directions should be added in the conclusion.

Author Response

Thank you for your contribution and efforts. We have provided a point-to-point reply in the attachment and sincerely hope that we can solve your questions.

Reviewer 2 Report

Comments and Suggestions for Authors

The authors in this paper have proposed a new deep reinforcement learning (DRL)-based strategy, i.e., DeepNR strategy, to enhance the energy efficiency and security performance of WSN. This paper is well written and organized, however, some major issues still need to be resolved so as to improve the quality of this paper, e.g.,

1) In the introduction, it is better to illustrate the background and novelty of this paper, so as to make readers understand the contribution more clearly.

2) More latest studies about DRL should be added and analyzed in the related work, e.g., Distributed Multi-Agent Reinforcement Learning for Cooperative Edge Caching in Internet of Vehicles, IEEE Transactions on Wireless Communications, 2023

3) More details about DRL should be added, e.g., which kind of DRL the authors adopts, and the model of DRL, the authors can refer to the above studies.

4) The authors need to introduce the reason why choose DRL, and compare the proposed DRL-based method with other existing DRL-based mehtods, so as to evaluate the performance of the proposed method.

5) Some typos and grammar mistakes exist in this paper, the authors need to check the whole manuscript carefully.

Comments on the Quality of English Language

Some typos and grammar mistakes exist in this paper, the authors need to check the whole manuscript carefully.

Author Response

(The authors gave the same response as above.)

Reviewer 3 Report

Comments and Suggestions for Authors

The paper describes a proposal for a DeepNR method designed to achieve efficient WSN transmission while ensuring security. This proposal, when it is compared with traditional decision-making can more accurately predict network status and external threats according to the authors.

The authors claim that their contribution consists of:

- new DeepNR strategy for improving the efficiency and security performance of WSN, which can perform adaptive learning based on the real-time state of the network and historical data,

- DRL-based adaptive encryption mechanism that dynamically adjusts 50 the encryption level according to the current network state and potential security 51 threats, thus achieving a trade-off between security and energy efficiency.

- The experiments verify the effectiveness and superiority of the proposed DeepNR compared with other conventional methods.

None of these claims is sufficiently described in the paper.

This paper has significant weaknesses and it is not ready for publication at this stage. Authors should primarily concentrate on the following major weaknesses:

1. Related work is not adequate and does not target the papers that strictly implement the usage of neural networks in building energy-efficient WSN routing protocols and safety. The presented related work describes only the appliance of neural networks and deep learning to various aspects of WSNs.

2.  The proposal is briefly described in Section 3.2. Strategy Optimization, on a little more than one page. In that form, the description of the proposal does not offer sufficient details for its understanding.

3.  There is no detailed description of the test environment. There is only a brief note about 500 MICAz nodes and CC2420 communication modules. It is not clear if it is an IEEE 802.15.4 or ZigBee network. Is there one or more PAN networks? There is no number of routing nodes (full function devices) and end nodes (reduced function devices). There is no data about the hierarchy levels in the WSN, etc.

4. In 4.1. Network Life cycle it is not clear how the proposal directly affects the life cycle benefits of this new approach,

5. The same is in 4.2. Data Throughput. There is no reference or explanation of how this proposal positively affects throughput,

6.  It is not clear in 4.3. Security Assessment how the proposal affects security improvements. Especially there is no explanation of how this proposal reduces the number of attacked nodes.

7.  The same issue is with energy distribution and balance as described in 4.4.

Following these comment I suggest authors to extend and fully revise the paper and it to resubmit it. 

Author Response

(The authors gave the same response as above.)

Round 2

Reviewer 1 Report

Comments and Suggestions for Authors

No further comments

Author Response

Thank you for your time and efforts.

Reviewer 2 Report

Comments and Suggestions for Authors

The current version can be accepted.

Comments on the Quality of English Language

Moderate editing of English language needed

Author Response

Thank you for your time and efforts. 

Reviewer 3 Report

Comments and Suggestions for Authors

The authors have successfully answered all my comments, except a couple of minor issues.

Regarding my comment No. 3 about the description of the test environment is expanded. This description still lacks the number of completed tests as well as the test duration. There is no discussion or comment on how the duration of the test can influence the results.  

The proposal description is expanded in Section 3.2. However, it is not clear what is the motivation of this proposal. The authors should explain it better.

Author Response

Thank you for your valuable comments and contributions to improve the quality of the manuscript. We have made a point-by-point response, which is attached. We hope that it will address your concerns.
